# A Soar-Based Space Exploration Algorithm for Mobile Robots

**DOI:** 10.3390/e24030426

**Published:** 2022-03-19

**Authors:** Fei Luo, Qin Zhou, Joel Fuentes, Weichao Ding, Chunhua Gu

**Affiliations:** 1School of Information Science and Engineering, East China University of Science and Technology, Shanghai 200237, China; luof@ecust.edu.cn (F.L.); y30190805@mail.ecust.edu.cn (Q.Z.); 2Department of Computer Science and Information Technologies, Universidad del Bío-Bío, Chillán 3780000, Chile; jfuentes@ubiobio.cl

**Keywords:** space exploration, cognitive computing, Soar, heuristic algorithms, reinforcement learning

## Abstract

Space exploration is a hot topic in the application field of mobile robots. Proposed solutions have included the frontier exploration algorithm, heuristic algorithms, and deep reinforcement learning. However, these methods cannot solve space exploration in time in a dynamic environment. This paper models the space exploration problem of mobile robots based on the decision-making process of the cognitive architecture of Soar, and three space exploration heuristic algorithms (HAs) are further proposed based on the model to improve the exploration speed of the robot. Experiments are carried out based on the Easter environment, and the results show that HAs have improved the exploration speed of the Easter robot at least 2.04 times of the original algorithm in Easter, verifying the effectiveness of the proposed robot space exploration strategy and the corresponding HAs.

## 1. Introduction

Space exploration refers to the robot exploration of an unknown environment with the goal of completing a specific task [1]. This problem and the use of mobile robots have been applied in multiple scenarios, such as searching for rescue operations [2], automating the self-driving vehicles [3], exploring the ground to clean up garbage, and so on.

To solve the space exploration problem of mobile robots, the most common method used is the frontier exploration algorithm. The robot explores the environment by moving to the boundaries of the known parts, and gradually expands the known map to the entire environment that needs to be explored, completing the space exploration task [4]. Heuristic algorithms have also been used in space exploration problems [5,6]. Some examples are the colony algorithm [7], the ant colony algorithm [8], and the particle swarm algorithm [9].

Reinforcement learning is another method used to solve the space exploration problem of mobile robots in unknown environments [10,11]. This is due to its characteristics that enable agents to have learning capabilities and recognition, e.g., through image processing. With reinforcement learning the robot’s state characteristics are defined, and the problem to be solved is limited by its dimensionality [12]. Therefore, deep reinforcement learning is utilized to solve the shortcomings of reinforcement learning. Through deep reinforcement learning, mobile robots can automatically learn state features and reduce the dimensionality of the problems through interaction with the environment [13,14,15].

A common weakness in the above methods is that the environment information needs to be explored or pretrained based on a static information [16]. It cannot be collected or updated during the space exploration for re-exploring and retraining, limiting the dynamic capabilities of the solutions.

In contrast, cognitive computing provides a mechanism to continuously learn knowledge in an interactive manner, e.g., in a dynamic environment with data changing continuously [17]. With the use of perceptual data [18], cognitive computing is utilized to simulate cognitive processes, such as thinking and memorizing operations [19]. By analyzing cognitive mechanism, cognitive agents can construct cognitive computing system, execute cognitive processes, and simulate human thinking processes, such as perception, attention, and memory [20].

Based on cognitive computing—which can process changing data as a human being can, and which is able to evaluate its cognitive memory through reinforcement learning—in this paper, we propose a model for mobile robot space exploration. In the construction of this model we use the cognitive architecture of Soar, where the long-term procedural memory is optimized. Furthermore, three heuristic algorithms based on the proposed space exploration model are presented to solve problems in unknown environments.

The contributions of this paper can be summarized as follows:Proposing a space exploration model based on the decision-making process of Soar for the cognitive robot agent. With the model, Soar is utilized to reason relevant tasks and objectives based on the problem space, so as to solve the space exploration problem.Proposing three heuristic space exploration algorithms as the novel planning knowledge based on the space exploration model, which are implemented in Easter to improve the space exploration speed of the mobile robot.

The rest of this paper is organized as follows. Related work is presented in Section 2. In Section 3, we briefly overview the architecture of Soar, and then detail its key components, such as its learning mechanism and decision process. Then, the modeling method of space exploration based on the Soar is introduced in Section 4. Experiments are carried out based on the Easter, a demo agent of Soar, in Section 5. Finally, conclusions are drawn in Section 6.

## 2. Related Work

Space exploration in unknown environments is one of the applications of mobile robots, which can effectively reduce the workload of the staff and even rescue the lives of victims. For example, in harsh urban search and rescue scenarios, mobile rescue robots can be deployed to help explore unknown chaotic environments while searching for trapped victims, reducing the workload for rescuers. To complete tasks of space exploration quickly and accurately, robots need to have the ability of complex obstacle avoidance and that of perception to real-time dynamic environments.

The earliest algorithm used to solve the single-robot space exploration problem is the frontier exploration algorithm. In [21], Erkan et al. combined the frontier exploration algorithm with the robotic operation system (ROS). They tested a real robot platform, and compared and analyzed the effects of different frontier target allocation methods through the total path length and total exploration time. Due to the slow speed of traditional frontier exploration algorithms, researchers committed to improve the frontier exploration algorithm to increase its exploration speed. For example, later proposals focused on combining the frontier-based exploration algorithm with the line-of-sight exploration algorithm [22] and adding additional parameters to improve the decision-making speed of frontier exploration algorithm [23].

On the other hand, some authors abstracted the robotic space exploration into the path planning or traveling salesman problems. For example, Pei-Cheng Song et al. [24] proposed an improved cuckoo search algorithm based on compact parallel technology for the 3D path planning problem, which effectively saves the memory of unmanned robots. Jeng-Shyang Pan et al. [25] proposed a hybrid differential evolution algorithm CIJADE that combines improved CIPDE (MCIPDE) and improved JADE (MJADE), and applied the algorithm to the path planning problem of unmanned aerial vehicles. Experimental results shows that the algorithm can effectively find the optimal or near-optimal flight path of the UAV. Based on it, researchers converted the single-robot space exploration problems into a multi-robot collaborative space exploration problems. The idea was to improve the speed of space exploration by bionic algorithms through multiple robots, such as particle swarm optimization (PSO) and whale optimization algorithm (WOA). In [26], the authors applied robotic particle swarm optimization (RPSO) to the robot space exploration problem, and the method of calculating the PSO global optimal parameters was utilized to maximize the robot’s exploration area. In [27], a new framework was proposed that integrates deterministic coordinated multi-robot exploration (CME) and meta-heuristic frequency correction WOA technology to simulate the predating behavior of whales for search and detection. In [28], researchers proposed a random detection algorithm and applied it to the WOA.

Haoran et al. [3] proposed a decision-making algorithm based on deep reinforcement learning, which learned exploration strategies from local maps through deep neural networks. Farzad et al. [2] combined the traditional frontier-based exploration algorithm with deep reinforcement learning, in order to enable the robot autonomously to explore the unknown chaotic environment.

Compared with other space exploration algorithms, the learning-based space exploration technology can explore the environment earlier, allowing a larger space area to be explored at the same time. However, tasks such as target searching and navigation of robots require a high-level spatial perception, reasoning, and planning abilities to adapt to the dynamically changing environment in practical application. Cognitive computing can enable robots with human-like cognitive level, which is an effective way to solve the problem mentioned above. FU Yan et al. [29] applied the ACT-R cognitive theory to establish a new knowledge representation and processing method for robots to improve the flexibility of natural language and promote the cooperation between humans and robots, which proved the effectiveness of applying cognitive computing to robot space exploration.

Soar is a cognitive computing architecture for constructing general intelligent systems developed by the Artificial Intelligence Laboratory of the University of Michigan [30]. It abstracts all goal-oriented behaviors as the selection and application of operators to a state. Long-term program knowledge is a key part of Soar architecture—without long-term program knowledge, Soar architecture cannot solve any problem. In this paper, we solve the space exploration problem of mobile robots in unknown environments by optimizing the long-term program memory of the Soar architecture.

## 3. Soar Architecture

### 3.1. Overview

The cognitive computing architecture of Soar abstracts all goal-oriented behaviors, such as the operator’s selection and application of a state. State refers to the current situation that needs to be solved, operator is used to transform from a state to another state, and the goal is the expectation of the problem that needs to be solved.

Soar architecture consists of four parts: long-term procedural knowledge, learning mechanism, working memory, and decision procedure. Its structure is illustrated in Figure 1. During the execution of Soar, it continuously applies the current operator and selects the next operator until it reaches the goal state [31]. The selection and application process of operators is shown in Figure 2. The current situation, including data from sensors, results of intermediate inferences proposed during decision procedure, active goals, and active operators is held in working memory. Long-term procedural knowledge specifies how to respond to different situations in working memory and can be thought of as the program for Soar. The Soar architecture is also able to adjust the operator’s choice knowledge according to a given reward function through a learning mechanism.

Soar stores the agent’s current environment information and long-term procedural knowledge possessed by the agent through memory. The current environment information of the agent is stored in the working memory, which mainly consists of data from sensors, intermediate inference results, targets, and operators. Long-term procedural knowledge defines how agents act in different situations and is stored in memory in the form of rules.

The working memory is composed of working memory elements (WMEs). Each WME contains specific information, such as “the length of A is 5”. Therein, “A” is called as the identifier, and WMEs which share this identifier are called objects in working memory. The “length” is an attribute of A, and each object has different attributes. Each attribute has a value associated with it, called as attribute value. Therefore, each WME is composed of a triple tuple, namely the identifier–attribute–attribute value.

Soar regards long-term procedural knowledge as rules stored in production memory, as shown in Figure 3. Each production has a set of conditions and a set of actions. If the production conditions match the working memory, the production is triggered and the operation is executed.

### 3.2. Learning Mechanism

Soar adopts the reinforcement learning (RL) mechanism to adjust operator selection knowledge based on a given reward function. In this section, the RL mechanism is detailed and explained how it is integrated with production memory, decision cycles, and state stacks.

Soar’s RL mechanism learns the Q value of the state–operator pair. Q values are stored as preferences which are numeric-indifferent, and these preferences are generated by RL rules. Each rule is represented as a production of Soar. Syntactically, each production [30] consists of the symbol “sp”, followed by an opening curly brace “{”, the production’s name (optional), the documentation string (optional), the production’s type (optional), comments (optional), the production’s conditions (also called the left side, or LHS), the symbol “–>” (literally: dash–dash–greater than), the production’s actions (also called as the right side, or RHS), and a closing curly brace “}”. Each element of a rule is separated by white space. Indentation and linefeeds are used by convention, but are not necessary. It is a RL rule if—and only if—the LHS of a rule checks the proposed operator and the RHS proposes to create a numeric-indifferent preference. Numeric-indifferent preference means the value of the proposed operator is an number.

For example, Algorithm 1 is an example of a RL rule because its LHS checks the proposed alternative operator <o>+ (the symbol “+” means that the operator <o> is a candidate for selection) and creates a numeric-indifferent operator <o> whose value is 1.5. The rule means: check whether there is an operator with attribute value (name = taskName, x = 3, y = 12). If it exists, then a numeric-independent operator is proposed.
**Algorithm 1** RL rule in the form of production.1:sp {2:$\;(state\langle s\rangle {\;}^{\wedge }name\;taskName{\;}^{\wedge }x\;3{\;}^{\wedge }y\;12\;$3:$\;{\;}^{\wedge }operator\;\langle o\rangle +)$4:$\;(\langle o\rangle {\;}^{\wedge }name\;move{\;}^{\wedge }direction\;left)$5:→6:$\;(\langle s\rangle {\;}^{\wedge }operator\;\langle o\rangle =1.5)$7:}

Soar’s RL mechanism is integrated with the decision cycle to update RL rules. Whenever the RL operator is selected, the value of the corresponding RL rule will be updated. There are two types of RL mechanism in Soar—Sarsa and Q-Learning. The algorithm presented in this paper adopts the Q-Learning learning mechanism. For Q-Learning, its updating formula is shown in Formula (1).
(1)$$\delta_{t}=\alpha \left(r_{t+1}+\gamma \max_{a\in A_{t+1}}Q\left(S_{t+1},a\right)-Q\left(S_{t},a_{t}\right)\right)$$
Therein, $Q\left(s_{t},a_{t}\right)$ represents the Q value of state S and operator *a* corresponding to the current decision cycle t, $A_{t+1}$ represents the set of operators proposed in the next decision cycle, $\max_{a\in A_{t+1}}Q\left(s_{t+1},a\right)$ indicates the maximum value of Q that the state can obtain in the next decision cycle, $r_{t+1}$ indicates the total reward value collected in the next decision cycle, while α and γ represent the learning rate and discount factor of the Q-Learning algorithm, respectively.

### 3.3. Decision Process

The execution of a Soar program goes through a series of decision cycles. Each decision cycle consists of five phases: the input phase, the proposal phase, the decision phase, the application phase, and the output phase. In the input phase, new sensor data is transferred to the working memory. In the proposal phase, all rules can be triggered in parallel and withdrawn as the state changes. Soar selects the next operator in the decision phase. Soar executes the operator selected in decision cycle phase, and acts on the external environment in the output phase.

Soar interacts with the external environment that allows it to receive input from the environment and affects the environment. To accomplish the interaction, Soar adopts the mechanism provided named as input-link and output-link. The input-link adds and deletes elements from working memory according to changes of the external environment. The output-link makes it act based on the external environment. The input-link is processed at the beginning of each execution process, and the output-link occurs at the end of each execution process.

## 4. Space Exploration Modeling

In order to solve the space exploration problem of the mobile robot through the cognitive architecture of Soar, this section constructs a model for space exploration of cognitive robot agent based on the decision-making process of Soar. Firstly, the external information perceived by the cognitive agent and the internal operation information are stored in the working memory of the agent. Then, the long-term procedure memory is set, and the processing mechanism of Soar is utilized to reason relevant tasks and objectives based on the problem space, so as to simulate human behaviors and be applied to the space exploration of mobile robots.

The model is depicted in Figure 4. Therein, long-term procedural knowledge mainly consists of task knowledge, feature knowledge, and planning knowledge. Task knowledge refers to the tasks that the mobile robot agent needs to complete, planning knowledge refers to the planning strategy representation of the robot agent to complete the task, and feature knowledge refers to the environment description of the problem that needs to be solved by Soar and the memory structure that needs to be initialized.

The space exploration process is described in Algorithm 2. Notice that the long-term procedure memory, which was introduced in Section III, is set in the form of productions, and it includes task knowledge, feature knowledge, and planning knowledge. Secondly, the working memory are initialized as shown in line 2, where the initial state $s_{0}$ and the goal state g are initialized. Then the cognitive robot agent will acquire the perception information from external environment and transmit data into working memory through the input-link as shown in line 3. Next, Soar will propose production according to the feature knowledge and select an action according to planning knowledge in long-term procedure memory, as shown in lines 4–5. Afterwards, Soar will apply the action by transmitting it through output-link to perform the action on the external environment. At the same time, the action will be appraised by RL mechanism with rewards as shown in line 8. In lines 9–13, the agent will be decided whether it has achieved the goal state. If it does, then it stops the iteration, otherwise it returns to line 3.
**Algorithm 2** Space exploration process based on Soar.1:Set the long-term procedure memory: task knowledge, feature knowledge and planning knowledge2:Initialize working memory, such as the initial state $s_{0}$ and the goal state g3:Perceive the data from the environment4:Transmit data through input-link5:Propose production according to the feature knowledge6:Select action according to planning knowledge7:Transmit the selected action a through output-link8:Appraise action by a RL mechanism9:Apply the selected action a to arrive next state $s_{-}$10:**if**$s_{-}$=g11:   stop12:**else if** s=$s_{-}$13:   return to line 3

## 5. Heuristic Space Exploration Algorithms

As described in Algorithm 2, the action of the space exploration process is determined by the planning knowledge. Although Soar can be utilized to construct general intelligent systems, the practical intelligent application system is still under research. One of the research systems is Easter (https://github.com/SoarGroup/Domains-Eaters-TankSoar/tree/master/agents/eaters, accessed on 16 January 2022), a mobile robot game created based on Soar, which is provided by the research team of Soar. We have applied the planning knowledge in Easter. To further improve the efficiency of the space exploration, three heuristic space exploration algorithms are proposed as the novel planning knowledge, including the left-hand algorithm, the right-hand algorithm, and the depth-first algorithm, which are also implemented in Easter.

### 5.1. Easter Overview

In Easter, eaters compete to consume food in a simple grid world, and the external environment is an L × L grid world, where L is the number of cells in the horizontal or vertical direction of the grid. The grid is surrounded by closed walls, and the intersection is differentiated as a normal food cell, a bonus food cell or an obstacle. The reward food cells and obstacles are randomly distributed in the intersections of the grid. Therein, the reward food cell has two types of reward values: $R_{normal}$ and $R_{bonus}$. The cell with $R_{normal}$ reward values is called as a normal food cell, and the cell with $R_{bonus}$ reward values is called as a bonus food cell. As it is shown in Figure 5, the bonus food cells are the red intersections, and the normal food cell are blue intersections, while the obstacles are black intersections.

The Easter robot agent starts from a random cell, and continues to move up, down, left, or right through space exploration strategy. When the robot agent moves in a cell without obstacles, it will receive corresponding rewards according to the type of cell. For example, it will receive $R_{bonus}$ rewards when it reaches the bonus food cell, while it will receive $R_{normal}$ rewards when it reaches the normal food cell. If all the food in the grid is obtained, then the robot stops exploring, or it continues the exploring process, which means that the goal of Easter is to eat all the food in the grid environment.

### 5.2. Space Exploration Algorithms

In the original Easter, a random exploration strategy is implemented, as can be seen in Algorithm 3. Therein, an Easter agent checks bonus food, normal food, and space, sequentially. If the number of the checked cell is more than one, then the Easter agent randomly chooses one of the checked cell and move to it. Otherwise, the agent moves to the unique cell. This section proposes an heuristic space exploration strategy to improve the exploration efficiency, as shown in Algorithm 4. The same as that in Algorithm 3, an Easter agent checks bonus food, normal food, and space, sequentially, in Algorithm 4. However, the Easter agent will move according to one of the proposed heuristic algorithms (HAs), such as the left-hand algorithm (Algorithm 5), the right-hand algorithm (Algorithm 6), or the depth-first algorithm (Algorithm 7), depicted as follows.

**Left-hand algorithm (LA)**: In the current state, the agent first judges whether there is an obstacle in the left cell. If not, then it moves to the left. Otherwise, it continues to judge if there is an obstacle in the front cell. If not, then it moves ahead, or it moves to the right.**Right-hand algorithm (RA)**: In the current state, the agent first judges whether there is an obstacle in the right cell. If not, then it moves to the right. Otherwise, it continues to judge if there is an obstacle in the front cell. If not, then it moves ahead, or it moves to the left.**Depth-first algorithm (DFA)**: In the current state, the agent first judges whether there is an obstacle in the front cell. If not, then it moves ahead. Otherwise, it continues to judge if there is an obstacle in the left cell. If not, then it moves to the left, or it moves to the right.

**Algorithm 3** Random space exploration.
1:Initial start state2:**if** There is $N_{bonus}$ bonus food around3:   **if** $N_{bonus}>1$4:      Move to one of bonus food points randomly5:   **else**6:      Move to the unique bonus food point7:**elseif** There is $N_{normal}$ normal food around8:   **if** $N_{normal}>1$9:      Move to one of normal food points randomly10:   **else**11:      Move to the unique normal food point12:
**else**
13:   **if** $N_{space}>1$14:      Move to one of $N_{space}$ space food points randomly15:   **else**16:      Move to the unique space food point


**Algorithm 4** Space exploration with heuristic algorithms.
1:Initial start state2:**if** This is $N_{bonus}$ bonus food around3:   **if** $N_{bonus}>1$4:      Move according to HA5:   **else**6:      Move to the unique bonus food point7:**elseif** There is $N_{normal}$ normal food around8:   **if** $N_{normal}>1$      Move according to HA9:   **else**10:      Move to the unique normal food point11:
**else**
12:   **if** $N_{space}>1$13:      Move according to HA14:   **else**15:      Move to the unique space food point


**Algorithm 5** Left-hand algorithm.
1:Initial start state2:**if** no obstacles on the left3:   Move left4:**elseif** no obstacles ahead5:   Move ahead6:
**else**
7:   Move right


**Algorithm 6** Right-hand algorithm.
1:Initial start state2:**if** no obstacles on the right3:   Move right4:**elseif** no obstacles ahead5:   Move ahead6:
**else**
7:   Move left


**Algorithm 7** Deep-first algorithm.
1:Initial start state2:**if** no obstacles ahead3:   Move ahead4:**elseif** no obstacles on the left5:   Move left6:
**else**
7:   Move right


## 6. Experiments and Performance Analysis

Experiments were carried out based on the Easter environment. Although the proposed model and HAs can also be implemented in other robot agents, currently only Easter is applicable for us, while the other practical intelligent robot systems are still under research. Therefore, we compared the proposed heuristic space exploration algorithms with the original one in Easter. In the remainder of this section, the experimental environment is first presented, followed by the comparative experiments. Analysis is further expanded based on the experimental results.

### 6.1. Configuration of the Experiments

Based on Easter, all the comparative algorithms were implemented in the Soar grammar, and their running environment was SoarJavaDebugger.

In order to enable the robot to recognize the attributes of the current state in the process of space exploration, the feature knowledge is defined as the alternative moving directions of the robot in the current state. Especially, it is represented in form of a triple tuple as (direction_one, action, direction_two). Therein, direction_one in the tuple is the source direction, action is the moving action of the agent, and direction_two is the destination direction. For example, in Figure 6, when the robot agent reaches the state S2 from the state S1, the feature knowledge is defined as (north, left, east), (north, ahead, north), (north, right, west). The meaning of the feature knowledge is described below.

**(north, left, east)** The agent is from S1 by moving **north** and has a state S2. If it continues moving **left**, then the destination direction is **east**.**(north, ahead, north)** The agent is from S1 by moving **north** and has a state S2. If it continues moving **ahead**, then the destination direction is **north**.**(north, right, west)** The agent is from S1 by moving **north** and has a state S2. If it continues moving **right**, then the destination direction is **west**.

Algorithms are compared by evaluating the exploration speed, *V*, which is defined as the number of explored effective grid intersections per unit step, as shown in Formula (2).
(2)$$V=\left(n_{t}-n_{o}\right)\times 10^{3}/n$$
$n_{o}$ represents the total number of obstacles in the grid, $n_{t}$ represents the total number of grid intersections, and n represents the number of steps required to obtain all the food in the grid world.

The speed rate *r* between the speed of the heuristic space exploration algorithms and that of the original space exploration algorithm in Easter is defined in Equation (Equation 3). Therein, $V_{Original}$ is the speed of the original space exploration algorithm in Easter, and $V_{alg}$ is the heuristic space exploration algorithm, where alg∈{LA,RA,DFA}.
(3)$$r=V_{alg}/V_{Original}$$

### 6.2. Results and Analysis

In Soar’s Q-Learning mechanism, the learning rate α is set as 0.3 and the discount factor γ is set as 0.9. Other experimental parameters are listed in Table 1, such as L, $R_{normal}$, $R_{bonus}$, and $R_{total}$. Therein, $R_{total}$ is the total reward of the whole grid world. The experimental results are shown in Table 2 ($R_{total}$ is set to 1500), Table 3 ($R_{total}$ is set to 3000), Table 4 ($R_{total}$ is set to 4500), Table 5 ($R_{total}$ is set to 6000).

Experimental results show that the exploration speed of HAs is always bigger than that of the original planning algorithm in Easter in different circumstances. Especially, the HAs can improve the exploration speed at least 2.04 times the original algorithm in Easter when $R_{total}$ = 1500, and at most 4.01 times the original algorithm in Easter when $R_{total}$ = 4500.

It can also be seen that none of the space exploration speed of HAs is larger than that of the other two algorithms. For example, when $R_{total}$ = 1500, $V_{LA}>V_{RA}$. In other circumstances, $V_{LA}<V_{RA}$.

Robustness experiments were further carried out with parameter values listed in Table 6, where L varies with the same values of $R_{bonus}$ and $R_{normal}$, and $R_{total}$ changes along with L. The results are shown in Table 2 (L = 15), Table 7 (L = 20), and Table 8 (L = 25). It can be seen that with the rise of L, the space exploration speeds up the original algorithm and HAs decreases. For example, when the grid size L rises from 15 to 20, the speed of LA decreases from 328.55 to 197.82. Although none of the HAs overwhelms others, the exploration speed of HAs is always faster than that of the original algorithm. Especially, when the grid size L is 20, the speed of RA is 8.36 times the original.

These results indicate that by applying the HAs in the planning knowledge, the space exploration speed of the mobile robot can be effectively improved.

## 7. Conclusions

This paper proposes a new space exploration model based on Soar that optimizes its long-term procedural memory to solve the space exploration problem of mobile robots in unknown environments. Three heuristic space exploration algorithms are also proposed to improve the exploration speed of the robot. The proposed model and the algorithms are implemented in the Easter environment. Experimental results show that when comparing the proposed heuristic algorithms with the original space exploration algorithm of Easter, the exploration speed is improved at least 2.04 times and at most 8.36 times. Results verify the effectiveness of the robot space exploration method based on the Soar architecture.

Currently, the proposed model and the corresponding HAs have only been implemented and compared in Easter, because most Soar-based intelligent robot agents are still under research. In the meantime, the proposed space exploration model still utilizes Soar’s native Q-Learning method. In the future, we will provide a general experimental environment to implement the model, and further improve the model with the improvement of the learning mechanism in Soar.

## Figures and Tables

**Figure 1 entropy-24-00426-f001:**
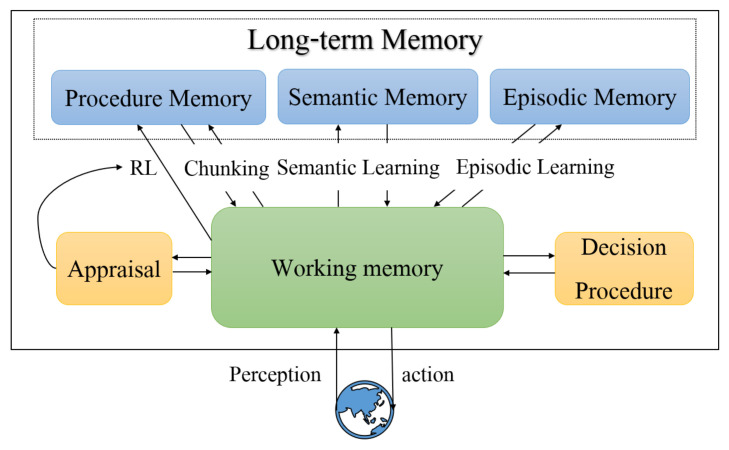
Soar architecture.

**Figure 2 entropy-24-00426-f002:**
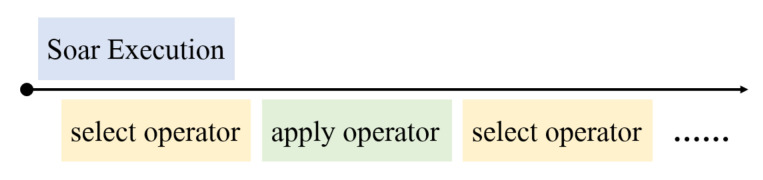
The execution process of Soar.

**Figure 3 entropy-24-00426-f003:**
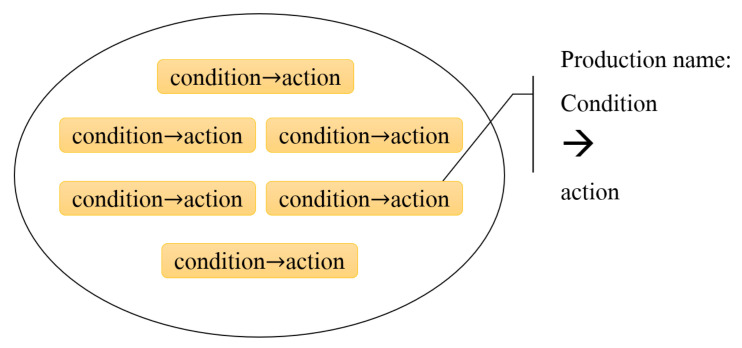
Productions in Soar.

**Figure 4 entropy-24-00426-f004:**
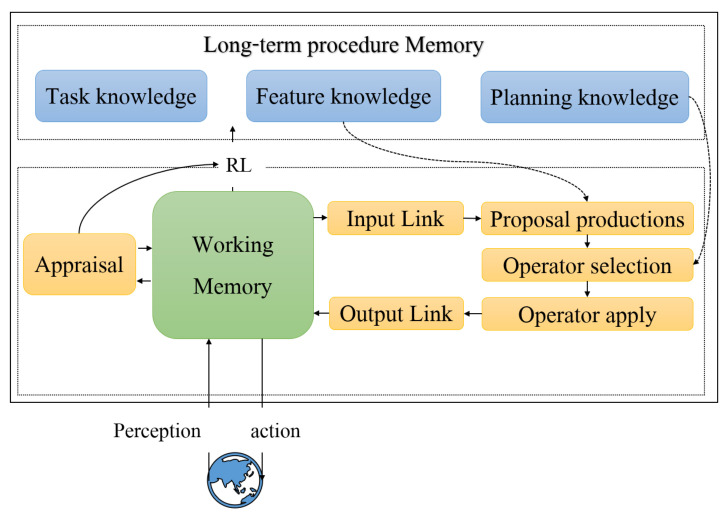
Soar-based space exploration model.

**Figure 5 entropy-24-00426-f005:**
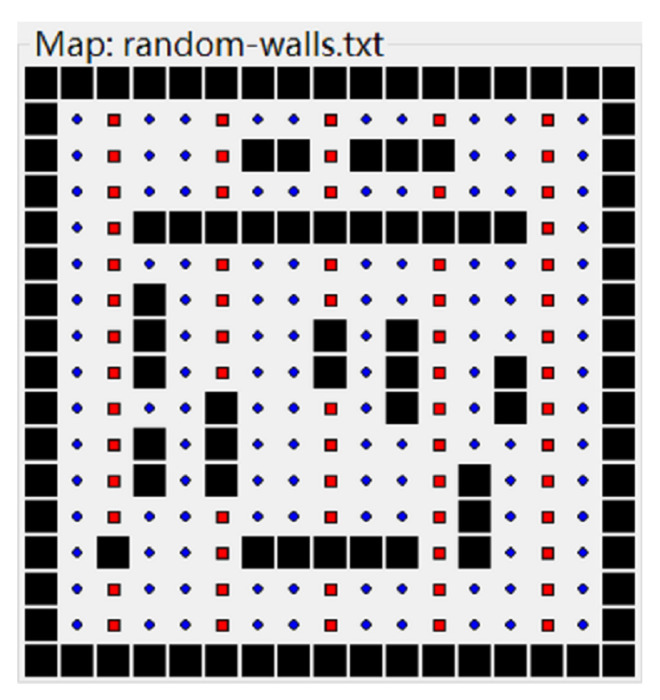
The food distribution in grid world.

**Figure 6 entropy-24-00426-f006:**
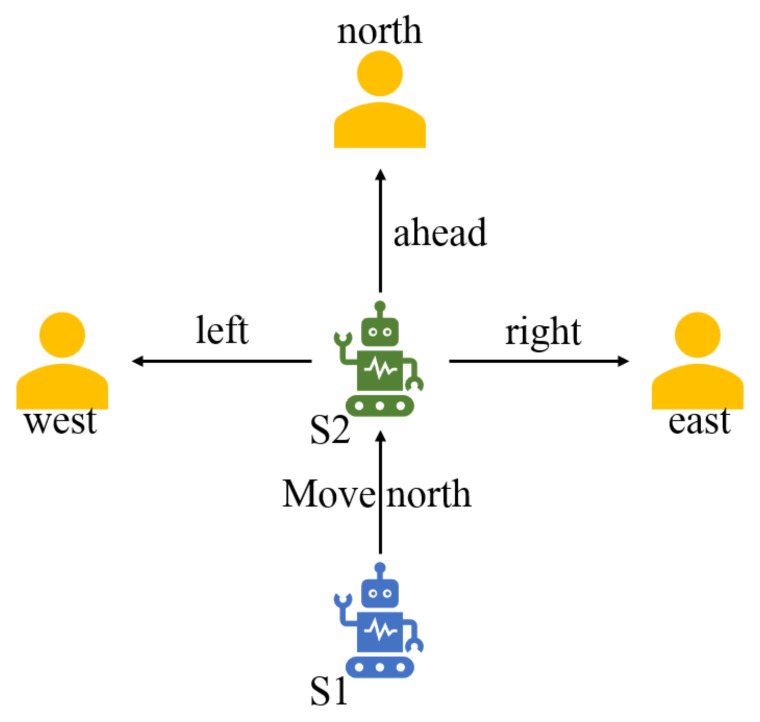
Mobile robot enters state S2 from state S1.

**Table 1 entropy-24-00426-t001:** Parameters setting.

Parameter	Value
L	15
$R_{total}$	1500, 3000, 4500, 6000
$R_{bonus}$	10
$R_{normal}$	5

**Table 2 entropy-24-00426-t002:** Exploration speed with $R_{total}$ = 1500.

Algorithm	$n_{o}$	$n_{t}$	n	V	r
Original	37	255	3496	62.36	/
LA	45	255	1653	127.04	2.04
RA	38	255	1629	133.21	2.14
DFA	42	255	862	**247.10**	**3.96**

**Table 3 entropy-24-00426-t003:** Exploration speed with $R_{total}$ = 3000.

Algorithm	$n_{o}$	$n_{t}$	n	V	r
Original	43	255	4261	49.75	/
LA	47	255	1311	158.65	3.19
RA	45	255	1210	173.55	3.49
DFA	45	255	1169	**179.64**	**3.61**

**Table 4 entropy-24-00426-t004:** Exploration speed with $R_{total}$ = 4500.

Algorithm	$n_{o}$	$n_{t}$	n	V	r
Original	48	255	3076	67.29	/
LA	40	255	1287	167.05	2.48
RA	41	255	1414	151.34	2.25
DFA	45	255	778	**269.92**	**4.01**

**Table 5 entropy-24-00426-t005:** Exploration speed with $R_{total}$ = 6000.

Algorithm	$n_{o}$	$n_{t}$	n	V	r
Original	53	255	4263	47.38	/
LA	43	255	1429	127.36	2.69
RA	49	255	1253	148.36	3.13
DFA	45	255	1346	**156.02**	3.29

**Table 6 entropy-24-00426-t006:** Robustness experimental parameters.

L	Parameter	Value
15	$R_{total}$	1500
	$R_{bonus}$	10
	$R_{normal}$	5
20	$R_{total}$	2700
	$R_{bonus}$	10
	$R_{normal}$	5
25	$R_{total}$	4125
	$R_{bonus}$	10
	$R_{normal}$	5

**Table 7 entropy-24-00426-t007:** Exploration speed with L = 20.

Algorithm	$n_{o}$	$n_{t}$	n	V	r
Original	84	400	10612	29.78	/
LA	91	400	1562	197.82	6.64
RA	85	400	1265	**249.01**	**8.36**
DFA	74	400	2184	149.27	5.01

**Table 8 entropy-24-00426-t008:** Exploration speed with L = 25.

Algorithm	$n_{o}$	$n_{t}$	n	V	r
Original	144	625	18453	26.07	/
LA	135	625	4115	119.08	4.57
RA	148	625	3028	**157.53**	**6.04**
DFA	139	625	4181	116.24	4.46

## Data Availability

Not applicable.

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
