# Peer review of "A Soar-Based Space Exploration Algorithm for Mobile Robots"

_entropy, 2022, doi:10.3390/e24030426_

Round 1

Reviewer 1 Report

See attached PDF file

Reviewer 2 Report

The authors propose a model for mobile robot space exploration. Three heuristic algorithms are presented to solve  the problem in unknown environment. As a all the paper is well written. I have some remarks, which can improve the paper:

  1. The figures are too small and unreadable
  2. Put in bold the best results in the tables
  3. There are several typos

Reviewer 3 Report

  • The left side of Algorithm 2 should indicate the number of lines corresponding to the code.
  • The format of 'if' and 'else' in Algorithms 3 and 4 should be aligned.
  • The paper has numerous grammatical errors and formatting errors. Please check the tenses of verbs and the use of conjunctions in sentences, as well as the formatting issues of paragraphs.
  • The meaning of n0 in Section 6.1 is unclear. Does it refer to the total number of obstacles in the grid, or the number of obstacles detected by the robot during the search?
  • The experimental data is too simple. Some comparison experiments with different grid sizes should also be considered.
  • Some recent works related to the path planning of the mobile robots, the authors may introduce those works as follows.

  1. “A Parallel Compact Cuckoo Search Algorithm for Three-Dimensional Path Planning”, Applied Soft Computing, Vol. 94, 2020, 106443, https://doi.org/10.1016/j.asoc.2020.106443
  2. “A hybrid differential evolution algorithm and its application in unmanned combat aerial vehicle path planning”, IEEE Access, 8, pp. 17691 - 17712, 2020

Round 2

Reviewer 3 Report

The authors have improved the quality of this paper, I recommend the acceptance of this paper.